# Special Solicitude: Religious Freedom at America's Public Universities

**William E. Thro**

Office of Legal Counsel, University of Kentucky, Lexington, KY 40506, USA; william.thro@uky.edu

**Abstract:** Rejecting the Obama Administration's argument that the First Amendment requires identical treatment for religious organizations and secular organizations, the Supreme Court held such a "result is hard to square with the text of the First Amendment itself, which gives special solicitude to the rights of religious organizations." (*Hosanna-Tabor*, 565 U.S. at 189). This "special solicitude" guarantees religious freedom from the government in all aspects of society, but particularly on public university campuses. At a minimum, religious expression and religious organizations must have equal rights with secular expression and secular organizations. In some instances, religious expression and religious expression may have greater rights. The Court's 2020 decisions in *Espinoza v. Montana Department of Revenue*, and *Our Lady of Guadalupe School v. Morrissey-Berru*, reinforce and expand the "special solicitude" of religion. Indeed, *Espinoza* and *Our Lady* have profound implications for student religious groups at America's public campuses. This article examines religious freedom at America's public universities. This article has three parts. First, it offers an overview of religious freedom prior to *Espinoza* and *Our Lady*. Second, it briefly discusses those two cases. Third, it explores the implications of those decisions on America's public campuses.

**Keywords:** religious freedom; higher education; constitutional law





## 1. Introduction

Rejecting the Obama Administration's argument that the First Amendment requires identical treatment for religious organizations and secular organizations, the Supreme Court held such a "result is hard to square with the text of the First Amendment itself, which gives special solicitude to the rights of religious organizations." (Laycock 2012, at 189). This "special solicitude" guarantees religious freedom from the government in all aspects of society, but particularly on public university campuses. At a minimum, religious expression and religious organizations must have equal rights with secular expression and secular organizations. In some instances, religious expression and religious expression may have greater rights.

The Court's 2020 decisions in *Espinoza v. Montana Department of Revenue*, and *Our Lady of Guadalupe School v. Morrissey-Berru*, reinforce and expand the "special solicitude" of religion. Indeed, *Espinoza* and *Our Lady* have profound implications for student religious groups at America's public campuses.

This article examines religious freedom at America's public universities. This article has three parts. First, it offers an overview of religious freedom prior to *Espinoza* and *Our Lady*. Second, it briefly discusses those two cases. Third, it explores the implications of those decisions on America's public campuses.

## 2. Overview of Religious Freedom at America's Public Universities

### 2.1. Student Religious Groups Have the Right to Recognition, Access, and Funding

There is "no doubt that the First Amendment rights of speech and association extend to the campuses of state universities." (*Widmar*, 454 U.S. at 269). A public university may not favor those groups that support the institution's views, and it may not penalize

those groups with which it disagrees (*Healey*, 408 U.S. at 187–88). Similarly, the Court has ruled that student religious groups are entitled to both access (*Widmar*, 454 U.S. at 267–70) and funding (*Rosenberger*, 515 U.S. at 831). Indeed, the practice of requiring students to pay mandatory fees that are then distributed to student groups is permissible only if the institution does not favor particular viewpoints (*Southworth*, 529 U.S. at 233–34). Quite simply, the "avowed purpose" for recognizing student groups is "to provide a forum in which students can exchange ideas." (*Widmar*, 454 U.S. at 272 n.10.). Thus, a group that holds racist, sexist, homophobic, anti-Semitic, or anti-Christian views is entitled to recognition, access to facilities, and funding. However, while the institution may not refuse recognition because of the student organization's viewpoint, the institution may require the organization to: (1) obey the campus rules; (2) refrain from disrupting classes; and (3) obey all applicable federal, state, and local laws (*Healy*, 408 U.S. at 185–86).

To be sure, this mandate of viewpoint neutrality toward student organizations does not mean that the university must compromise its own viewpoints. While the institution must accommodate the viewpoints of all student groups, "students and faculty are free to associate to voice their disapproval of the [student organization's] message." (*Rumsfeld*, 547 U.S. at 69–70). If one finds a particular viewpoint disagreeable, the solution is to promote an alternative viewpoint, not to suppress the disagreeable viewpoint.

However, officials often refuse to fund activities of student religious groups that, in the judgment of the institution, are "worship activities" or proselytizing while expressly allowing funding for virtually identical activities by secular groups (Roman Catholic Found., 578 F. Supp. 2d at 1134–35). For example, administrators might allow the French Club to buy bread and wine for its functions but deny the Roman Catholic Club's request to buy bread and wine. They might subsidize the community outreach activities of political groups or advocacy groups but refuse to subsidize the evangelism activities of religious groups (Roman Catholic Found.,578 F. Supp. 2d at 1134–36).

*2.2. Student Religious Groups Have Limited Rights of Association*

The right to express a particular viewpoint necessarily includes the right to associate with others who share that view. "An individual's freedom to speak, to worship, and to petition the government for the redress of grievances could not be vigorously protected from interference by the State unless a correlative freedom to engage in group effort toward those ends were not also guaranteed." (*Roberts*, 468 U.S. at 622). "This right is crucial in preventing the majority from imposing its views on groups that would rather express other, perhaps unpopular, ideas." (*Dale*, 530 U.S. at 647–48) "If the government were free to restrict individuals' ability to join together and speak, it could essentially silence views that the First Amendment is intended to protect." (*Rumsfeld*, 547 U.S. at 69–70) This freedom of association "is not reserved for advocacy groups. However, to come within its ambit, a group must engage in some form of expression, whether it be public or private." (*Dale*, 530 U.S. at 648).

"Freedom of association . . . plainly presupposes a freedom not to associate." (*Roberts*, 468 U.S. at 623). "Freedom of association would prove an empty guarantee if associations could not limit control over their decisions to those who share the interests and persuasions that underlie the association's being." (*Democratic Party*, 450 U.S. at 122 n. 22). "The forced inclusion of an unwanted person in a group infringes the group's freedom of expressive association if the presence of that person affects in a significant way the group's ability to advocate public or private viewpoints." (*Dale*, 530 U.S. at 648).

Therefore, government may intrude on the freedom of association only "by regulations adopted to serve compelling state interests, unrelated to the suppression of ideas that cannot be achieved through means significantly less restrictive of associational freedoms." (*Roberts*, 468 U.S. at 623). Courts must "examine whether or not the application of the state law would impose any 'serious burden' on the organization's rights of expressive association." (*Dale*, 530 U.S. at 685). Judges "give deference to an association's assertions regarding the nature of its expression" and "to an association's view of what would impair

its expression." (*Id.* at 653). It is not necessary for the organization's core purpose to be expressive or for all members to agree with all aspects of the message (*Id.* at 655). Under this framework, the Court has upheld statutes requiring civic organizations to admit women, (*Roberts*, 468 U.S. at 623–27) but has allowed both parade organizers, (*Hurley*, 515 U.S. at 572–73) and the Boy Scouts to exclude homosexuals (*Dale*, 530 U.S. at 655–60). The cases have turned on whether the "the enforcement of these [policies]" would "materially interfere with the ideas that the organization sought to express." (*Id.* at 657).

However, with respect to student religious groups at public universities, different rules often apply. In *Christian Legal Society v. Martinez*, a sharply divided Supreme Court upheld—as a matter of federal constitutional law—policies at public institutions requiring student groups to admit "all comers." Under this precedent, as a condition of becoming recognized student organizations, a status affording them such benefits as access to campus facilities and some funding, religious groups must admit "all comers," including those who disagree with their deeply held religious beliefs and values (*Christian Legal Society*, 561 U.S. at 668). Put another way, in *Christian Legal Society*, the Supreme Court declared that the government, through university officials, could force faith-based groups to choose between compromising their religious values and receiving benefits that other student groups receive as a matter of constitutional right. While the government "surely could not demand that all Christian groups admit members who believe that Jesus was merely human," (*Id*. at 731 (Alito, J., joined by Roberts, C.J., Scalia and Thomas, J.J., dissenting)), government "may impose these very same requirements on students who wish to participate in a forum that is designed to foster the expression of diverse viewpoints." (*Id*.). As Professor Paulsen notes, the "holding is a fundamental negation of the right of Christian campus groups to freedom of speech, to freedom of association, and to the collective free exercise of religion—a First Amendment disaster trifecta." (Paulsen 2012, at 284).

Moreover, at some institutions, while secular student groups may exclude those who disagree with their views, religious organizations were required to refrain from what they described as religious discrimination as they sought to preserve their faith-based identities (e.g., Alpha Delta). For example, the Young Democrats may have excluded Republicans, but Evangelical Christian Clubs could not have denied membership to atheists (*Id*. at 800–01).

Although nothing in the Court's opinion limits *Christian Legal Society* to a particular context, the reality is the case arose in an unusual factual situation. Although most public institutions allow student groups to exclude those who disagree with the group's goals or do not share the group's interests, *Christian Legal Society* involved a policy forbidding any student organization from discriminating for any reason. Under this "all-comers policy," the Young Democrats had to allow Republicans to join; the Vegetarian Society had to include carnivores; and the Chess Club had to allow members who would prefer to play checkers. If an institution allows some student political organizations or student special interest organizations to exclude those who do not share the group's ideology, interests or values, then it will be difficult to justify forcing student religious groups to admit nonbelievers.

### 2.3. Public Universities May Not Force People of Faith to Violate Their Beliefs

Certain professional groups, such as psychological counselors or social workers, impose ethical requirements on those who are part of the profession (Demitchell et al. 2013, at 304–305). Yet, adhering to those ethical requirements may require people of faith to violate their religious beliefs (Laycock 2014, at 872–73). As part of training students to enter the profession, public university faculty may insist students conform to the profession's ethics and ignore their faith convictions (Demitchell et al. 2013, at 305).

When confronted with the issue, the lower federal appellate courts have agreed that religious students must conform to the profession's ethics (*Keeton*, 664 F.3d at 874), but have insisted that students with religious objections be treated the same as secular students (*Ward*, 667 F.3d at, 735–38). Nevertheless, the insistence that religious students conform to

professional ethics that are antithetical to their beliefs can chill such students from even entering the profession. (Laycock 2014, at 872–73).

A 2018 decision expands the right of religious students to dissent. The Court held California's legislature violated the Constitution by requiring professionals, "to inform women how they can obtain state-subsidized abortions." (Nat'l Inst., 138 S. Ct. at 2371). The constitutional challenge involved a group of professionals who were opposed to abortion and who actively tried to persuade women form pursuing abortion (*Id.* at 2370) By compelling the professionals to speak a particular message, the government was "altering the content" of the professional's speech (*Id.* at 2371). Most significantly, the Court rejected the notion, embraced by some of the courts of appeal, that strict scrutiny does not apply to content-based regulation of "professional speech." (*Id.* at 2371–75). This aspect of the holding broadens the freedom of speech for professionals and aspiring professionals speaking in their professional context (Mattox 2018).

Although faculty and administrator may look to compel an affirmation of certain views, *Nat'l Inst. of Family and Life Advocates* reaffirms the State "must not be allowed to force persons to express a message contrary to their deepest convictions. Freedom of speech secures freedom of thought and belief." (Nat'l Inst., 138 S. Ct. at 2379 (Kennedy, joined by Roberts, C.J., Thomas, Alito, and Gorsuch, J.J., concurring). Similarly, to the extent that public university administrators look to punish speech of aspiring professionals for not adhering to professional norms, *Nat'l Inst. of Family and Life Advocates* rejection of lesser scrutiny for "professional speech" precludes such actions (Mattox 2018). In sum, all members of the university community are free from compulsion. Just as speech from professionals will be the same as speech from ordinary citizens for constitutional purposes, speech from students who aspire to a particular profession must be treated the same as speech from ordinary students.

### 3. The Decisions Expanding Religious Freedom

#### 3.1. Espinoza

In *Trinity Lutheran Church v. Comer*, the Court announced a new constitutional rule: Except where such an action would violate the Establishment Clause, the Free Exercise Clause prohibits constitutional actors from conferring or denying a benefit solely because of an individual's or entity's religious exercise (Trinity Lutheran, 137 S. Ct. 2025). *Trinity Lutheran* arose when officials of a faith-based preschool and daycare center applied to Missouri's Department of Natural Resources hoping to take part in its Scrap Tire Program to acquire material to better protect the children in its care by installing a new playground surface. The Program, which offered a limited number of reimbursement grants to re-duce the volume of used tires in landfills and dump sites, supplied funds to nonprofit organizations to acquire safe materials for playground surfaces made from recycled tires. However, four justices joined a footnote suggesting the result was limited to the context of the program at issue (*Id.* at 2024 n. 3. (Roberts, C.J., joined by Kennedy, Alito, and Kagan, J.J., announcing the judgment of the Court)).

In *Espinoza*, the Court expanded *Trinity Lutheran* to all contexts by invalidating a Montana state constitutional provision that prohibiting aid to religious organizations (Russo and Thro 2020). Consequently, if a State chooses to subsidize private education, then the State cannot exclude religious schools from the subsidy solely because they are religious (Espinoza, 140 S. Ct. at 2261). *Espinoza* effectively prohibits government at all levels from treating religious organizations worse than it treats secular organizations (*Id.* at 2256). While the Court previously recognized "some space for legislative action neither compelled by the Free Exercise Clause nor prohibited by the Establishment Clause," (*Cutter*, 544 U.S. at 719), the Court narrowed that space (Espinoza, 140 S. Ct. at 2260–61).

By invalidating the "Blaine Amendments" in various State Constitutions, *Espinoza* ensures that government treats all religious people with equal dignity. Born in the anti-Catholic bigotry of the late Nineteenth Century, the Blaine Amendments treated religious organizations differently from similarly situated secular organizations (*Id.* at 2267–74

(Alito, J., concurring)). The implicit message of the Blaine Amendment was that religious organizations were unwelcome in public life and people of faith, particularly Catholics, were second class citizens to the extent they sought to express faith. *Espinoza* ends this stain on our constitutional character.

After *Espinoza*, if government supplies funding or benefits for private secular entities, then it must also supply funding or benefits for religious entities. People of faith and their organizations will no longer be second class citizens in our Constitutional Republic. This rule alone makes *Espinoza* a landmark. In a society where no faith commands a majority, the Court acknowledged that while a sizable number of people embrace no faith and a vocal minority is hostile to any faith, it is essential that government does not treat people of faith and their organizations as second-class citizens.

### 3.2. *Our Lady*

In *Hosanna-Tabor Evangelical Lutheran Church and School v. EEOC* recognized that the Religion Clauses guarantees "a religious organization's freedom to select its own [leaders]." (*Hosanna-Tabor*, 565 U.S. at 189). "By imposing an unwanted minister, the state infringes the Free Exercise Clause, which protects a religious group's right to shape its own faith and mission through its appointments." (*Id.* at 188). "According the state the power to determine which individuals will minister to the faithful also violates the Establishment Clause, which prohibits government involvement in such ecclesiastical decisions." (*Id.* at 188–89).

*Hosanna-Tabor* establishes that religious groups have a right of absolute discretion to determine who their leaders will be. Logically, if an organization can restrict its leadership to those who adhere to the faith and basic principles, then the organization ought to be able to impose a similar requirement on membership. Consequently, the necessary inference of *Hosanna-Tabor* is that religious organizations, through the Religion Clauses, have greater associational freedoms than their secular counterparts do (Laycock 2012, at 855).

In *Our Lady*, the Court expanded *Hosanna-Tabor's* "ministerial exception," to include teachers who were not ordained ministers, had no religious training, and who taught secular subjects (Our Lady, 140 S. Ct. at 2055). In doing so, the Court noted that a "variety of factors may be important" in determining if the ministerial exception applies (*Id.* at 2063). *Our Lady* held government must acknowledge a religious organization's "autonomy with respect to internal management decisions that are essential to the institution's central mission. Additionally, a component of this autonomy is the selection of the individuals who play certain key roles." (*Id.* at 2060) "When a school with a religious mission entrusts a teacher with the responsibility of educating and forming students in the faith," the First Amendment prohibits "judicial intervention into disputes between the school and the teacher." (*Id.* at 2069). While the Court gave significant deference to the religious organization's judgement, Justice Thomas, joined by Justice Gorsuch, would give absolute deference: "The Religion Clauses require civil courts to defer to religious organizations' good-faith claims that a certain employee's position is 'ministerial.'" (*Id.* at 2069–70 (Thomas, J., joined by Gorsuch, concurring)).

## 4. The Potential Implications of Espinoza and Our Lady

### 4.1. *Redefining the Constitutional Space Between the Establishment and Free Exercise Clauses?*

A written constitution establishes the parameters of the government, but also limits the government (*Afroyim*). As such, it limits the discretion of constitutional actors to pursue a particular end by a particular means. These limitations on sovereign discretion take two forms—prohibitions and requirements (Thro 2016). While Americans are familiar with the idea of constitutional provisions as prohibitions, they are less familiar with the notion of constitutional provisions that impose requirements on government to act in a particular way (Zackin 2013, at 36–47).

When constitutional actors do something that is prohibited or do not do what is required, the judiciary must intervene and force them to comply. Yet, in the space between



what the Constitutions need and what the Constitutions prohibit, elected legislative and executive actors have absolute discretion to pursue whatever policy goals they wish. Within this constitutional space, legislative and executive actors can choose how to remedy constitutional violations (*Milliken*). As any "ruling of unconstitutionality frustrates the intent of the elected representatives of the people," (Regan, 468 U.S. at 652 (White, J., joined by Rehnquist, C.J. and O'Connor, J., announcing the judgment of the Court)), the judiciary cannot force the legislative and executive actors to choose a particular course when other courses are equally constitutional (*Horne*, 557 U.S. at 450).

To be sure, there will be times—particularly involving constitutional prohibitions—when the constitutional space is small or even nonexistent. Historically, the space between what the Establishment Clause prohibits/requires and what the Free Exercise Clause prohibits/requires has been quite broad. Indeed, prior to *Trinity Lutheran*, constitutional actors routinely conferred or denied a benefit solely because of an individual's or entity's religious exercise (Thro and Russo 2017). *Trinity Lutheran* narrowed the constitutional space and *Espinoza* narrowed it further (Russo and Thro 2020).

In his concurrence in *Espinoza*, Justice Thomas, joined by Justice Gorsuch, suggested it was necessary to redefine the constitutional space between the Free Exercise and Establishment Clauses. First, the "modern interpretation" of the Establishment Clause is too broad and often conflicts with the Free Exercise Clause. Constitutional actors, at all levels, often rely on the Establishment Clause as a justification for infringing on the free exercise of the rights of individuals and groups. For example, officials at public universities unsuccessfully argued that the Establishment Clause compelled them to deny access or funding to student religious groups while extending these benefits to secular groups. In effect, the modern view of the Establishment Clause, diminishes the vitality of the Free Exercise rights such that it becomes the "lowest rung" on the constitutional ladder (Espinoza, 140 S. Ct. at 2267 (Thomas, J., joined by Gorsuch, J. concurring).

Second, Justice Thomas observed that the "modern view, which presumes that States must remain both completely separate from and virtually silent on matters of religion to comply with the Establishment Clause, is fundamentally incorrect." (*Id.* at 2264). As Justice Thomas noted on various occasions, the original public meaning of the Establishment Clause was not to create an individual right but, rather, was intended to prevent the National Government from establishing a religion or interfering with the States' efforts to maintain their own established churches at the state level (*Id.* at 2264). In sum, under the Original Understanding, the Establishment Clause does not set up a limitation on the States.

However, even if the Establishment Clause does limit the States, "it would only protect against an 'establishment' of religion as understood at the founding, i.e., 'coercion of religious orthodoxy and of financial support by force of law and threat of penalty.'" (*Id.*) Instead of focusing on keeping a "separation of church and state," the Court may focus on the "freedom from a religious establishment." (Hamburger 2002) For example, Thomas believes that the judiciary should never rely on the Establishment Clause to reach such questions as whether a display of a creche was sufficiently secular or whether a governing body could solemnize its meetings with a prayer.

If the Court were to adopt Justice Thomas' view, the constitutional space between the Establishment Clause and Free Exercise Clause would be redefined. The reach of the Establishment Clause would be scaled back. This would eliminate the tension between the Religion Clauses. It would also allow government, if it wished, to supply accommodations that are beyond the requirements/prohibitions of the Free Exercise Clause. The result would be a path toward greater religious freedom, a reaffirmation of a foundational purpose of early American colonization, and, most significantly, a way to achieve a "Confident Pluralism." (Inazu 2016).

Three points here are crucial. First, if individual rights under the Establishment Clause are limited to avoiding coercion, the government will have no justification for treating people of faith worse than secular citizens. Second, while America had multiple

beginnings (Woodard 2011), the desire to worship God as they saw fit was vital to the origin of Maryland, Rhode Island, and Utah. (Sutton 2018, at 17). In fact, the Mayflower Compact among the pilgrims fleeing religious persecution is a foundation of our constitutional order (Kelly et al. 1991, at 8–10). Third, because Americans "lack agreement about the purpose of our country, the nature of the common good, and the meaning of human flourishing," (Keller and Inazu 2020, at xvi), it is essential that we develop the grace to address our differences through humility, patience, and tolerance (*Id.* at xvii–xix), While the Opinion of the Court establishes a broad principle of equality between the religious and secular organizations, the Thomas concurrence points the way toward a society where all people of faith and people of no faith can thrive.

### 4.2. Undermining Christian Legal Society

Of course, *Christian Legal Society* remains controlling constitutional precedent unless, or until, it is explicitly overruled (*Agostini*, 521 U.S. at 237–38). Even so, as I have explained previously, later decisions undermine *Christian Legal Society* (Thro 2013, Thro 2014). In particular, the holdings of *Hosanna-Tabor* and *Our Lady* are in conflict with *Christian Legal Society*.

To explain, *Hosanna-Tabor* and *Our Lady* establish the principle that religious groups have a right of religious autonomy—absolute discretion to select their leaders. While *Hosanna-Tabor* involved an incorporated church rather than an unincorporated student religious group, there is no reason to think that the rights of church members or student group members depend upon the organizational form. Logically, if organizations can restrict their leadership to those who adhere to their faiths, then they ought to be able to establish similar requirements for membership. This is the opposite result to *Christian Legal Society*.

### 4.3. Public Universities Must Provide Equal Treatment to Religious Organizations and Secular Organizations

After *Espinoza*, America's public universities must provide equal treatment to both religious organizations and secular organizations. This is so for two reasons.

First, since *Espinoza* invalidated all the Blaine Amendments, public universities may no longer rely on provisions of their State Constitutions to justify treating student religious organizations differently than secular organizations. By narrowing, if not abolishing, the State's ability to make religious policy, the Court effectively promoted religious freedom.

Second, because *Trinity Lutheran* and *Espinoza* prohibits governmental officials from focusing on the religious identity of student organizations, policies and/or practices treating them differently than their secular counterparts must cease. In deciding whether to fund refreshments or outreach activities, the religious nature of organization is, or should be, rendered meaningless. Simply put, in the wake of *Trinity Lutheran* and *Our Lady*, in assessing the validity of membership policies, officials at public institutions cannot apply different standards to student religious groups.

### 5. Conclusions

Many Americans have "a deep faith that requires them to do things passing legislative majorities might find unseemly or uncouth." (Espinoza, 140 S. Ct. at 2277 (Gorsuch, J. concurring)). The Religion Clauses mandate "special solicitude" toward people of faith as they express their beliefs and come together for religious purposes. Over a period of two decades—from *Widmar* in 1981 to *Rosenberger* in 1995 to *Southworth* in 2000—the Court mandated that public institutions must treat religious speech and organizations in the same manner as secular speech in organizations. Yet, religious individuals still had to violate their conscience and religious organizations had to admit nonbelievers. The Court's 2020 decisions in *Espinoza* and *Our Lady* further expand religious freedom. This expansion has significant implications for America's public universities. As a result of *Espinoza* and *Our Lady*, all public campuses are "wide enough" (Miranda 2015) for everyone—atheist and

believer, secular and sacred, clergy and laity, Muslim and Jew, Protestant and Catholic
(Russo et al. 2020).

**Funding:** This research received no external funding.

**Institutional Review Board Statement:** Not applicable.

**Informed Consent Statement:** Not applicable.

**Data Availability Statement:** Not applicable.

**Conflicts of Interest:** The author declares no conflict of interest.

## References

### Cases

*Afroyim v. Rusk*, 387 U.S. 253, 257 (1967)
*Agostini v. Felton*, 521 U.S. 203 (1997)
*Alpha Delta Chi-Delta Chapter v. Reed*, 648 F.3d 790 (9th Cir. 2011)
*Bd. of Regents of Univ. of Wisconsin Sys. v. Southworth*, 529 U.S. 217 (2000)
*Boy Scouts of Am. v. Dale*, 530 U.S. 640 (2000)
*Christian Legal Society v. Martinez*, 561 U.S. 661 (2010)
*Cutter v. Wilkinson*, 544 U.S. 709 (2005)
*Democratic Party of U.S. v. Wisconsin ex rel. LaFollette*, 450 U.S. 107 (1981)
*Espinoza v. Montana Department of Revenue*, 140 S. Ct. 2246 (2020)
*Healy v. James*, 408 U.S. 169 (1972)
*Horne v. Flores*, 557 U.S. 443 (2009)
*Hosanna-Tabor Evangelical Lutheran Church & Sch. v. E.E.O.C.*, 565 U.S. 171 (2012)
*Hurley v. Irish-Am. Gay, Lesbian and Bisexual Grp. of Boston, Inc.*, 515 U.S. 557 (1995)
*Keeton v. Anderson-Wiley*, 664 F.3d 865 (11th Cir. 2011).
*Milliken v. Bradley*, 433 U.S. 267 (1977)
*Nat'l Inst. of Family & Life Advocates v. Becerra*, 138 S. Ct. 2361(2018)
*Our Lady of Guadalupe School v. Morrissey-Berru*, 140 S. Ct. 2049 (2020)
*Regan v. Time, Inc.*, 468 U.S. 641(1984)
*Roberts v. U.S. Jaycees*, 468 U.S. 609 (1984)
*Roman Catholic Found., UW-Madison, Inc. v. Regents of Univ. of Wisconsin Sys.*, 578 F. Supp. 2d 1121 (W.D. Wis. 2008)
*Rosenberger v. Rector and Visitors of the University of Virginia*, 515 U.S. 819 (1995)
*Rumsfeld v. Forum for Academic & Inst'l Rights*, 547 U.S. 47 (2006)
*Trinity Lutheran Church v. Comer*, 137 S. Ct. 2012 (2017)
*Ward v. Polite*, 667 F.3d 727 (6th Cir. 2012)
*Widmar v. Vincent*, 454 U.S. 263 (1981)

### Other Sources

Demitchell, Todd A., David T. Herbert, and Loan T. Phan. 2013. The University Curriculum, and The Constitution: Personal Beliefs and Professional Ethics in Gradate School Counseling Programs. *The Journal of College and University Law* 39: 303.
Hamburger, Phillip. 2002. Separation of Church & State. Available online: https://www.hup.harvard.edu/catalog.php?isbn=9780674013742 (accessed on 23 February 2021).
Inazu, John D. 2016. *Confident Pluralism*. Chicago: University of Chicago Press. Available online: https://press.uchicago.edu/ucp/books/book/chicago/C/bo23291067.html (accessed on 23 February 2021).
Keller, Timothy, and John Inazu. 2020. *Uncommon Ground: Living Faithfully in a World of Difference*. Edinburgh: Thomas Nelson.
Kelly, Alfred H., Winfred A. Harbison, and Herman Belz. 1991. *The American Constitution: Its Origins and Development 8–10*, 7th ed. New York: W. W. Norton & Company.
Laycock, Douglas. 2012. Hosanna-Tabor and The Ministerial Exception. Available online: https://www.harvard-jlpp.com/wp-content/uploads/sites/21/2013/10/35_3_839_Laycock.pdf (accessed on 23 February 2021).
Laycock, Douglas. 2014. Religious Liberty and the Culture Wars. *University of Illinois Law Review* 2014: 839–80.
Mattox, Casey. 2018. *Three New Supreme Court Decisions Protect Speech on Campus*. National Review On-Line. Available online: https://www.nationalreview.com/2018/08/supreme-court-decisions-clarify-campus-free-speech-protections/ (accessed on 23 February 2021).

Miranda, Lin-Manuel. 2015. *The World Was Wide Enough (from the Musical Hamilton)*. London: Kobalt Music Publishing Ltd.

Paulsen, Michael Stokes. 2012. Disaster: The Worst Religious Freedom Case in Fifty Years. Available online: https://www.regent.edu/acad/schlaw/student_life/studentorgs/lawreview/docs/issues/v24n2/02Paulsenvol.24.2.pdf (accessed on 23 February 2021).

Russo, Charles J., and William E. Thro. 2020. *Born of Bigotry, Died in Religious Liberty: The Supreme Court Ends the Blaine Amendments in Empowering Parental Choice*. Atlanta: Emory University Canopy Forum on the Interactions of Law & Religion.

Russo, Charles J., William E. Thro, and Allan G. Osborne Jr. 2020. Reaffirming the First Freedom: The Implications of Espinoza v. Montana Department of Revenue and Our Lady of Guadalupe School v. Morrissey-Berru. *Religion & Education* 47: 86–105.

Sutton, Jeffery. 2018. *51 Imperfect Solutions: States and the Making of American Constitutional Law*. Oxford: Oxford University Press.

Thro, William E. 2013. Undermining Christian Legal Society v. Martinez. Available online: https://papers.ssrn.com/sol3/papers.cfm?abstract_id=2319468 (accessed on 23 February 2021).

Thro, William E. 2014. The Limits of Christian Legal Society. Available online: https://heinonline.org/HOL/LandingPage?handle=hein.journals/denovo2014&div=8&id=&page= (accessed on 23 February 2021).

Thro, William E. 2016. No Clash of Constitutional Values: Respecting Freedom & Equality in Public University Sexual Assault Cases. Available online: https://www.regent.edu/acad/schlaw/student_life/studentorgs/lawreview/docs/issues/v28n2/8_Thro_vol_28_2.pdf (accessed on 23 February 2021).

Thro, William E., and Charles J. Russo. 2017. Odious to the Constitution: The Educational Implications of Trinity Lutheran Church v. Comer. Available online: https://papers.ssrn.com/sol3/papers.cfm?abstract_id=3033967 (accessed on 23 February 2021).

Woodard, Colin. 2011. *American Nations: A History of the Eleven Regional Cultures of North America*. London: Penguin Books.

Zackin, Emily. 2013. *Looking for Rights in All the Wrong Places: Why State Constitutions Contain America's Positive Rights*. Princeton: Princeton University Press.