# Peer review of "Special Solicitude: Religious Freedom at America’s Public Universities"

_laws, 2011_

Round 1

Reviewer 1 Report

This is an excellent piece of scholarship, drawing together recent Supreme Court jurisprudence and applying it to the public university setting.  Increasingly, this is where the principal issues of free exercise are arising.  This piece will make an important contribution to those debates.

Author Response

Thank you for your comments.

Reviewer 2 Report

Please see attached word file

Author Response

Thank you for your comments.

Reviewer 3 Report

The author write about different cases ehich are connected with different  judicial doctrines and standards of review, and he does not investigate in depth them.

About the Guadalupe case the author should develop the judicial track about church autonomy (Hosanna-Tabor) and investigate comparable and distinctive aspects of the case he is focusing on.

About Espinoza: Since the Everson case, there is a long evolution of US Supreme Court relating to access to public funding of religious organizations that should be analyzed.

The conclusions should be improved: What are the real new aspects of the 2020 case law? What is their impact on the interpretation of the First Amendment?

Author Response

Thank you for your valuable comments in expanding the piece well beyond its current boundaries and to explore some fascinating issues. While I have explored some of those issues in other publications and I likely will write a longer piece for another publication that further develops my thoughts.

Round 2

Reviewer 3 Report

As far as I'm concerned the paper I reviewed can be published in the present form.